# Multilevel Spiral Axicon for High-Order Bessel–Gauss Beams Generation

**DOI:** 10.3390/nano13030579

**Published:** 2023-01-31

**Authors:** Rebeca Tudor, George Andrei Bulzan, Mihai Kusko, Cristian Kusko, Viorel Avramescu, Dan Vasilache, Raluca Gavrila

**Affiliations:** 1National Institute for Research and Development in Microtechnologies IMT, 077190 Bucharest, Romania; 2Faculty of Physics, University of Bucharest, 405 Atomistilor Street, 077125 Magurele, Romania

**Keywords:** OAM, spiral axicon, divergence-free beams, fused silica optics, propagation invariance

## Abstract

This paper presents an efficient method to generate high-order Bessel–Gauss beams carrying orbital angular momentum (OAM) by using a thin and compact optical element such as a multilevel spiral axicon. This approach represents an excellent alternative for diffraction-free OAM beam generation instead of complex methods based on a doublet formed by a physical spiral phase plate and zero-order axicon, phase holograms loaded on spatial light modulators (SLMs), or the interferometric method. Here, we present the fabrication process for axicons with 16 and 32 levels, characterized by high mode conversion efficiency and good transmission for visible light (λ = 633 nm wavelength). The Bessel vortex states generated with the proposed diffractive optical elements (DOEs) can be exploited as a very useful resource for optical and quantum communication in free-space channels or in optical fibers.

## 1. Introduction

Light is a reliable candidate for optical communication through its degrees of freedom, which facilitate the increase in the security and capacity channel of the transmitted information. Polarization, wavelength, amplitude, frequency, etc., all make a significant contribution to the encoding of information in large amounts and secure modes [1]. Forty years have passed since the seminal paper of Allen et al. [2], who introduced the concept of one special degree of freedom of light, namely orbital angular momentum (OAM). Both industry and researchers have focused on the study of OAM, which facilitates, in theory, the construction of an alphabet with an infinite number of symbols transmitted as orthogonal states [3,4,5,6,7,8,9].

Optical vortices (OVs) are special beams carrying OAM and can be generated with cylindrical lenses, spiral phase plates, fork-like computer-generated holograms, q-plates, plasmonic structures, and integrated optical circuits [1,4,9]. Most of these optical elements generate diverging OVs, which involves complicated detection in cases of long-distance communication. One solution for this is to use nondivergent OVs, called high-order Bessel–Gauss beams, which exhibit annihilation reconstruction properties and stability in atmospheric turbulences [5,8]. One can generate diffraction-free OVs with axicons [10,11,12,13,14], either by loading a high-order phase hologram on a spatial light modulator [15,16,17], amplitude binary masks [15], or shaping a Laguerre–Gauss beam with a refractive axicon—a doublet of a real spiral phase plate and a zero-order refractive axicon (with continuous profile) [18], Pancharatnam–Berry phase elements [18], a digital micromirrors device [19], circular Dammann gratings [20], Mach–Zehnder interferometer [21], cylindrical waveguides [22], conformal transformation elements (for OAM detection) [23,24,25], or metasurfaces [26,27].

Optical elements such as axicons transform a Gaussian beam into a quasi-Bessel beam [28] or Bessel–Gauss [29,30], which is a nondiffractive optical beam at a specific propagation distance, depending on the geometrical parameters of these beam shapers. After the propagation at the distance *z_max_*, the lateral energy is not available anymore for the self-reconstruction of the central lobes, so the wavefront of the beam is distorted. Bessel beams are nondiffractive optical beams associated with a Bessel function of order, *m*, in paraxial approximation [13]. 

Spiral axicons are aspherical elements generating Bessel–Gauss beams that present singular geometry with a helical profile [13]. A specific parameter for the axicons is the focal length, *z_max_ = w/[(n − 1) δ],* where *w* represents the waist of the incident beam, *n* the refractive index of the axicon, *δ = arctan(q/R),* and *q* and *R* are the cone’s height and radius, respectively. Axicons are used in various applications such as imaging, optical tweezers or optical spanners, beam shaping, material processing, and optical and quantum communication in free space or in fiber [13].

A practical and efficient approach for the generation of Bessel–Gauss beams carrying orbital angular momentum is to use a Fresnel/spiral axicon, also called a multilevel spiral axicon. Two transmission functions, corresponding to the zero-order axicon, and a spiral phase plate can be multiplexed together in order to generate a single and compact beam shaper with higher efficiency, which accomplishes two functions: it generates OAM states and makes them nondivergent at a specific propagation distance.

In this work, the procedure of fabricating axicons and spiral axicons consists of the application of microfabrication technologies, more specifically, photolithography followed by chemical etching processes in order to realize optical/micro-optical components with free-form optical surfaces and various diameters of the order of millimeters. While for free-form optical surfaces, various sophisticated techniques, such as single-point diamond-turning machining [31,32], molding [33], 3D lithography (mask-less) [34], or electron beam lithography [35], photolithography on a grey-level photoresist [36], or Laser-Assisted Wet Etching [37], are applied, they present several challenges in terms of complexity, costs, and the diameters of the components to be fabricated. Microfabrication techniques represent a mature, well-established technology [31], a standard in micro/nanoelectronics presenting the advantages of low cost, mass production, and flexibility in terms of the wafer diameters. It is the aim of this work to develop and establish a technological flow using simple technological processes for the realization of optical components with aspherical surfaces, specifically, axicon and spiral axicons.

## 2. Numerical Methods

In this part of the paper, we present the numerical studies concerning the optimal parameters of the axicons (radius, height, focal length) for the design and fabrication process. We performed numerical calculations and simulations based on the Kirchhoff diffraction integral in order to establish the optimal parameters for the technological design of aspherical phase elements such as axicons or spiral axicons. The diffraction of a Gaussian beam on these optical elements generates Bessel–Gauss beams of zero-order or high-order, respectively. The phase factor corresponding to the axicons depends on the wavelength of the incident beam, *λ*; *q*, the height of the axicon; *R*, the radius of the axicon; and *n*, the refractive index of the substrate. This phase factor is introduced in the diffraction integral as:(1)$$\Phi =\frac{2\pi }{\lambda }\frac{q}{R}\left(n-1\right)\sqrt{{x^{\prime }}^{2}+{y^{\prime }}^{2}}$$

The geometrical parameters of the axicons used for the simulations were the following. For the zeroth-order axicons, the parameter *δ* = *q*/*R*, the ratio between the height and the radius, is 0.0043, while, for the higher-order axicons, *δ* = 0.0012. These parameters were used in the design and fabrication of the axicons. The beam’s wavelength is *λ* = 635 nm; the waist of the incident beam is *w* = 1700 μm; and the propagation distances in the free-space range ranged from z = 0.5 m to z = 1.5 m. One can observe a numerical axicon of zero order with a discrete profile (32 levels) in Figure 1a. The intensity of a Gaussian beam diffracted by this axicon is illustrated in Figure 1b at z = 75 cm.

The spiral axicon with a discrete profile (32 levels) generating a Bessel–Gauss beam of order m=1 is represented in Figure 2a. The intensity distribution for z = 75 cm is illustrated in Figure 2b.

Figure 2c shows a numerical axicon of high-order m = 4 with a discrete profile (32 levels), while the intensity distribution of a Bessel beam of order four generated after the diffraction of a Gaussian beam on the spiral axicon is illustrated in Figure 2d.

We also plotted the intensity distribution of Bessel–Gauss beams of zero order in a cross-section for the geometrical parameters at z = 50 cm and z = 75 cm, (Figure 3a) and the one of the Bessel–Gauss beams of the first order at z = 50 cm and z = 75 cm, (Figure 3 b). Both figures illustrate divergence-free behavior along propagation distance.

One can verify that the intensity distribution is according to the one which is characteristic of the Bessel functions. The intensity distribution is proportional to the squared Bessel functions:(2)$$I\approx {\left[J_{n}\left(k_{r}x\right)\right]}^{2}$$
where *n* represents the order of the Bessel function, and the radial component of the wavevector is defined in the equation:(3)$$k_{r}=\frac{2\pi }{\lambda }\left(n-1\right)\delta$$
where *n* is the refractive index. The position *x*, corresponding to the first minimum for each order (*m* = 0 and *m* = 1) in the transverse direction considering the argument for which the Bessel function becomes zero, is:(4)$$x_{0}=\frac{2.405}{k_{r}}\;\;\;\;\;\;$$
(5)$$x_{1}=\frac{3.8317}{k_{r}}\;\;\;\;\;$$

The numerical results for these minimum positions are similar to the values obtained in the simulations. The value of the first minimum is the same in the profile of the Bessel functions, both for *m* = 0 and *m* = 1 order. The first minimum in the case of zero order is *x*_0_ = 121 μm, and, in the case of *m* = 1, is *x*_1_ = 681 μm for both z = 50 cm and z = 75 cm, which demonstrates the propagation invariance property of the Bessel beams. One can observe from the cross-section intensity distribution presented in Figure 3 how the Bessel function profile is kept along the propagation axis.

## 3. Results

### 3.1. Design for the Photolithographic Masks for Axicons

Fresnel axicons are designed based on the Fresnel lens principle by preserving only the surface profile of the kinoform element, which is responsible for the optical power generation, so the final DOE is thinner and lighter. For this reason, the diffractive axicon which enables the generation of zero-order Bessel–Gauss beams is designed by “transforming” the continuous profile of a refractive axicon of radius *R* and height *q*, illustrated in Figure 4a in discrete structures which introduces modulo 2π phase shifts in the wavefront, being of two wavelengths’ height (Figure 4b), This new thin axicon is characterized by the propagation distance *z_max_*, for which the Bessel–Gauss beam is diffraction-free (Figure 4c).

The Fresnel axicon presents discretized structures, illustrated in Figure 4b, and has the height *D* (discretized in *L* levels), which depends only on the incident wavelength *λ* and refractive index of the axicon *n*:(6)$$D=\frac{\lambda }{n-1}$$

The height compensation from *q* to *D* is given by a number of *T* periods:(7)$$T=\frac{q}{D}$$

The radius, *r*, of the Fresnel axicon depends on the parameters *D*, *R*, and *q*, while *δ* is the base angle of the refractive axicon:(8)$$r=\frac{D*R}{q}$$
(9)$$\delta =\frac{q}{R}$$

The radius, *r_i_*, corresponding to each of the *m* = 5 photolithographic masks is given by the equation:
(10)$$r_{i}=\frac{D\;*\;R}{q}\;*\;\frac{1}{\;L}\;*\;2^{m-i}$$

The maximum distance, *z_max_*, on which the Bessel–Gauss beam is diffraction-free depends on *R*, *q*, *n*, and *δ*, according to the equation:(11)$$z_{max}=\frac{R^{2}}{q*\left(n-1\right)}=\frac{R}{\delta *\left(n-1\right)}\;\;\;$$

Equation (11) is valid when the whole axicon is illuminated; for the case where the waist, w, is smaller than *R*, the numerator is replaced with the product between *R* and w.

The conversion efficiency of an axicon can be enhanced by increasing the number of levels *L* of DOE, according to the formula [38]:(12)$$\eta 1={\left[\frac{sin\left(\frac{\pi }{L}\right)}{\frac{\pi }{L}}\right]}^{2}$$

We have chosen silica as the substrate for the fabrication of DOE to be fused, taking into account the optical quality of this material, which guarantees good transmission at the visible wavelength λ = 635 nm. We have designed five photolithographic masks for the fabrication process in case of a multilevel element. The five masks permitted the fabrication of an axicon with 32 levels. In the case of a zero-order axicon, the width of the first ring, which corresponds to the first mask, is a1 = 160 μm (Figure 5a); for the second mask, the ring’s width is a2 = 80 μm (Figure 5b); for the third mask, it is a3 = 40 μm (Figure 5c); for the fourth mask, it is a4 = 20 μm (Figure 5d); and, for the fifth mask, the ring width is a5 = 10 μm (Figure 5e).

In the case of the axicon of order *m* = 1, with *R* = 1165, μm the photolithographic masks are presented below in Figure 6. The subfigures from (a) to (e) have the same scale bar—a = 145 μm). 

The height of the axicon is given by:(13)$$H=\frac{\lambda }{n-1}=1385.12\;\mathrm{nm}$$

In the photolithographic processes, a positive photoresist of HPR 504 at 3000 rpm was used. For each of the five photolithographic masks presented in Figure 5 and Figure 6, wet etching was performed in order to achieve particular etching depths: for the first photolithographic mask—h_1_ = 696 nm, for the second photolithographic mask—h_2_ = 348 nm, for the third photolithographic mask—h_3_ = 174 nm, for the fourth photolithographic mask—h_4_ = 87 nm, and for the fifth photolithographic mask—h_5_ = 44 nm.

### 3.2. Technological Flow for the Fabrication of the Multilevel Axicons

In general, one can fabricate diffractive optical elements on a fused silica substrate by using reactive ion etching, RIE [39,40], or wet etching based on HF/HNO_3_ [41,42] or KOH solution [43,44], or a laser writing axicon written in the polymer [44], EBL [45]. These fused silica etchants present good results in terms of high optical quality and low roughness. The use of KOH solution presents the advantage of safety in comparison to the acid HF solution. The disadvantage of KOH solution for wet etching consists of the expense for masking—the use of Cr-Au and high temperatures for high etching rates. We investigate a low-cost solution for wet etching with the major advantage of a short fabrication time, which is convenient for rapid fabrication, taking into consideration all five wet etching processes needed for 32-level axicon fabrication. This solution also confers precise control of the etch depth. The multilevel axicons working in transmission are fabricated by wet etching fused silica in hydrofluoric acid and ammonium fluoride, HF:H_4_FN (1:6), T = 22–23 °C, and an etching rate of 100 nm/min. 

We fabricated multilevel axicons (with 16 and 32 levels—see flow chart in Figure 7) by using, as the substrate, a 4-inch fused silica wafer which involves four or five photolithographic processes, respectively, followed by wet etching.

At the beginning of the fabrication, the wafer was cleaned in acetone. After that, the surface of the wafer was prepared for photolithography with a thermal treatment at 150 °C for 15 min. HPR 504 resist (FujiFilm) was applied on the wafer by spin-coating at 3000 rpm and prebaking at 90 °C for 1 min on a hotplate. Then, the resist was exposed to ultraviolet light for 3 s through the first photolithographic mask, M1, and developed in 10 s. A thermal treatment followed at 110 °C for 30 min in the oven. The wet etching of the fused silica wafer was performed in hydrofluoric acid and ammonium fluoride, HF:H_4_FN (1:6), at T = 22–23 °C, for 7 min, at the etching rate of 100 nm/min. The photoresist was removed in acetone, and the wafer was cleaned in Piranha solution (H_2_SO_4_+H_2_O_2_: 660 mL + 220 mL). After that, the surface of the wafer was prepared for the photolithographic process corresponding to the second mask by conducting a thermal treatment at 140 °C for 2 h in plasma. The process of the exposure of the mask and development for M2 was performed in the same conditions as in the case of M1. The wet etching had the same parameters for the solution, with an etching time of 3 min and 35 s. The pattern of mask M3 was transferred on the fused silica wafer in the same manner as mask M2 was, with an etching time of 1 min and 45 s. During the fourth photolithographic process, corresponding to mask M4, there were some challenges regarding the adhesion of the photoresist to the fused silica substrate. We found a solution to overcome this issue by using HMDS (hexamethyldisilazane-controlled) as an adhesion promoter. Before the photolithographic process, the M4 wafer was in a vacuum for 30 min. After that, the promoter HMDS was introduced in the vacuum for 1 min. The photolithographic process for mask M4 was performed based on the same parameters as in the case of mask M3, followed by a wet etching process in HF:H_4_FN (1:6) solution for 50 s. The photolithographic process in the case of mask M5 was similar to the one performed for M4, and the wet process in HF:H_4_FN (1:6) solution was conducted for 50 s.

The axicons were characterized structurally with a white light interferometer (WLI), atomic force microscope (AFM), and mechanical profilometer. The functional characterization of the fabricated axicons consisted of recording the intensity distribution given by the diffraction of a HeNe laser on the axicons at various propagation distances, but also the wavefront distribution in the Mach–Zehnder interferometer.

### 3.3. Structural Characterization

In order to increase the diffraction efficiency [46], the number of levels was increased, so the axicons with 16 levels were fabricated by using M1, M2, M3, and M4 in four photolithographic processes followed by wet etching. Figure 8 presents the cross-section profile performed with the mechanical profilometer on the zero-order axicon with 16 levels. Considering the reference value for the fourth mask, a step difference of 87 nm established in the design phase, one can observe that the absolute error is approximately 0.2 nm (relative error 0.22%). The conversion efficiency, in this case, is 98.72%.

One can increase the diffraction efficiency of the axicon from 16 levels to 32 levels. In this case, the axicons were fabricated by using M1, M2, M3, M4, and M5 in five photolithographic processes followed by wet etching. Figure 9 presents the cross-section profile of the zero-order axicon with 32 levels performed with the mechanical profilometer. The height difference between two consecutive levels is around 46 nm. Considering as reference value a step difference of 44 nm established in the design phase for the fifth mask, one can approximate the absolute error at 2 nm (relative error 4.5%).

In the case of a spiral axicon of order *m* = 4 with 16 levels fabricated in fused silica, the measured height difference between the minimum and maximum levels is 1.34 µm.Since the theoretical value is 1.385 µm, the relative error after fabrication is 3.24%. The differences between two consecutive levels have an average value of 89.47 nm. For this reason, the relative error after fabrication is 2.83%, taking into consideration the reference value established in the design phase for the fourth mask, a step difference of 87 nm.

Figure 10 represents the tridimensional WLI profile for the zero-order axicon (Figure 10a, spiral axicon *m* = 4 (Figure 10b), fabricated in fused silica with 32 levels, while Figure 10c illustrates a cross-section profile of the axicon from Figure 10a. The measured height difference between the minimum and maximum levels is 1.3824 µm. Since the theoretical value is calculated at 1.3851 µm, the fabricated DOE has a relative error of 0.21%. Considering as the reference value a step difference of 44 nm between two consecutive levels established in the design phase for the fifth mask, the average measured value is 44.31 nm, which generates a relative error of 0.7% for both the spiral and zero-order axicons. 

The AFM image with the cross-section profile of the 32-level spiral axicon fabricated in the fused silica has a 44.7 nm step, as illustrated in Figure 11a, while the surface smoothness illustrated in Figure 11b has a roughness of 0.8 nm for a 7 μm^2^ area.

### 3.4. Functional Characterization

The functional characterization of the fabricated multilevel axicons consisted of recording the intensity and wavefront distribution of the Bessel–Gauss beams generated in a Mach–Zehnder interferometer, depicted in Figure 12. A Gaussian beam emitted by a HeNe laser with a 633 nm wavelength and P = 7 mW power is attenuated with a variable attenuator in order to decrease the power of the incident beam. The diameter of the beam was adjusted with an iris. The axicon (zero-order or high order) was put on one arm of the Mach–Zehnder interferometer, while the other arm of the interferometer rests unchanged. The diffraction of the Gaussian beam on the axicon permits the generation of the Bessel–Gauss beam, whose intensity distribution is recorded with a CCD camera. The interference between a Gaussian beam and a high-order Bessel–Gauss beam (with an annular intensity distribution carrying OAM) permits the generation of a fork-like pattern representing the wavefront distribution of the high-order Bessel–Gauss beam. The number of the fork’s fringes indicates the OAM state of the beam.

The intensity distribution measured at the distance z = 40 cm for the first-order Bessel–Gauss beam with *m* = 1 generated with the axicon with *L* = 32 levels is illustrated in Figure 13a, while the intensity distribution for the higher-order Bessel–Gauss beam with *m* = 4 generated with spiral axicon with *L* = 32 levels is presented in Figure 13b. The annular distribution of the wavefront, a feature of the beams with helical wavefronts, as well as the concentric rings, which characterize a Bessel beam, can be observed. For the field generated by the first-order axicon, the measured diameter of the first ring is approximately 0.8 mm, while, for the fourth-order axicon, the diameter is approximately 1.6 mm. By comparing Figure 13 with Figure 1 and Figure 2, one can see that the experimental data are in semi-quantitative agreement with the numerical calculations.

In order to put in evidence the helicoidal character of the wavefront, and to determine the topological charge of the beams generated by higher-order axicons, the interference patterns obtained with the aforementioned Mach–Zehnder interferometer are shown in Figure 14. The observed interference fringes correspond with the standard fork patterns, whereas the dislocation present in the center of the beam demonstrates the phase singularity and the helicoidal structure of the wavefronts. For the beams generated by the first-order axicon (*m* = ±1), the difference between the number of the fringes present above the dislocation and the number of the fringes below the dislocation is ±1, while, for the beams generated by the fourth-order axicon, the difference is 4. These results show that the beams diffracted by the fabricated components present the characteristics of Bessel–Gauss beams.

## 4. Divergence-Free Behavior

In order to show the divergence-free behavior of the Bessel–Gauss beams generated by the fabricated 32-level axicons, we recorded with a CCD camera the intensity distribution at several propagation distances. The results for the zero-order Bessel–Gauss beams are shown in Figure 15, where one can observe similar intensity distributions for three different propagation distances of 30 cm, 35 cm, and 42 cm. The intensities for the experimental figures are normalized to 1.

For a more quantitative analysis, we overlaid the cross-sections of the 2D plots from Figure 15 with the intensity of an ideal zeroth-order Bessel beam, |J_0_(k_r_x)|^2^, where the transverse momentum, k_r_, is determined by the geometrical parameters of the fabricated axicon. The results displayed in Figure 16 show that, within experimental errors, the maxima of the measured intensities are in the same positions and are in good agreement with the zeroes and maxima of the ideal Bessel beam. 

A similar analysis was done for the first-order axicon. As in the case of the zeroth Bessel–Gauss beam, the experimental results for the propagation distances of 30 cm, 40 cm, and 50 cm, shown in Figure 17, indicate that the diameters of the first ring are practically the same. 

The cross-sections obtained from the 2D plots are represented in Figure 18. As in the previous case, there is a good overlap between the experimental curves, as well as a good correspondence between the intensity maxima of the experimental data and the intensity of the ideal Bessel beam, |J_1_(k_r_x)|^2^.

## 5. Conclusions

In this paper, we presented the fabrication of thin and compact, aspherical optical elements (zero-order and high-order spiral axicons) with advanced functionalities in optical and quantum communication, but also optical signal processing, by using microfabrication techniques. Photolithography and chemical etching are the principal fabrication processes that allow for high reproducibility and high optical quality for multilevel axicon realization. We investigated a low-cost solution for wet etching, with the major advantage of a short fabrication time. The multilevel axicons working in transmission were fabricated by wet etching fused silica in hydrofluoric acid and ammonium fluoride, HF:H_4_FN (1:6), T = 22–23 °C, and an etching rate of 100 nm/min. We succeeded in fabricating zero-order and spiral axicons with 16 and 32 levels, characterized by high mode conversion efficiency η = 98.72% and η = 99.68%, respectively, and good transmission for visible light (λ=633 nm wavelength). Although in this work the apertures of the axicons are in the range of millimeters, the maturity of the photolithographic techniques allows the realization of optical elements, specifically axicons, with diameters comparable with the wafer size. This will give the possibility to generate nondiffractive Bessel–Gauss beams with long propagation distances. 

The Bessel–Gauss vortex states generated with the proposed DOEs can be exploited as a very useful resource for optical and quantum communication in free-space channels or in optical fiber.

## Figures and Tables

**Figure 1 nanomaterials-13-00579-f001:**
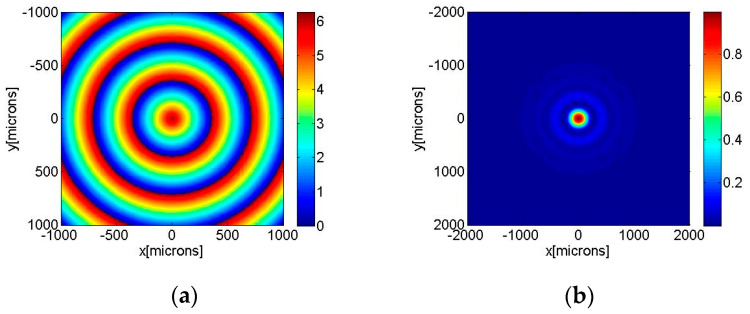
Zeroth-order axicon. (**a**) Numerical axicon of zero order with discrete profile (32 levels), (**b**) intensity of a Bessel–Gauss beam of zero order generated after the diffraction of a Gaussian beam on the axicon for z = 75 cm.

**Figure 2 nanomaterials-13-00579-f002:**
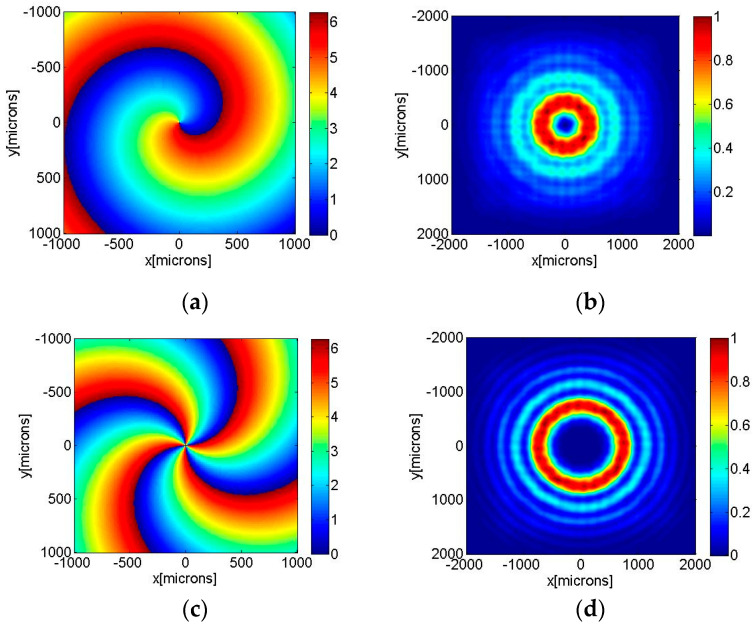
Higher-order axicons with discrete profile (32 levels): (**a**) numerical axicon of first order, *m* = 1, (**b**) intensity distribution of a Bessel–Gauss beam of order *m* = 1 generated after the diffraction of a Gaussian beam on the spiral axicon for z = 75 cm, (**c**) numerical axicon of order *m* = 4, (**d**) intensity distribution of a Bessel–Gauss beam of order m = 4 generated after the diffraction of a Gaussian beam on the spiral axicon for z = 75 cm.

**Figure 3 nanomaterials-13-00579-f003:**
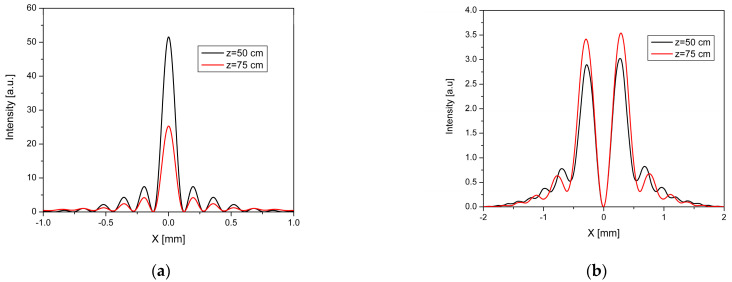
Intensity distribution in cross-section generated by: (**a**) zero-order axicon at z = 50 cm (black line) and z = 75 cm (red line), (**b**) spiral first-order axicon at z = 50 cm (black line) and z = 75 cm (red line).

**Figure 4 nanomaterials-13-00579-f004:**
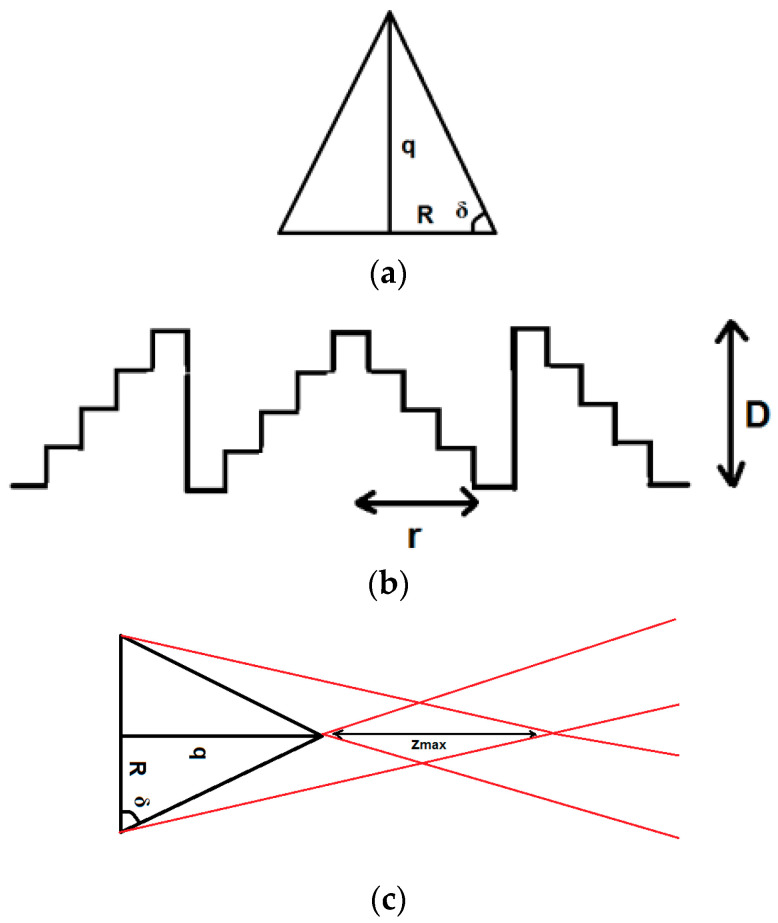
(**a**) Refractive axicon with continuous profile; (**b**) DOE—diffractive (Fresnel) axicon; (**c**) *z_max_*—propagation distance for which the Bessel–Gauss beam is diffraction-free.

**Figure 5 nanomaterials-13-00579-f005:**
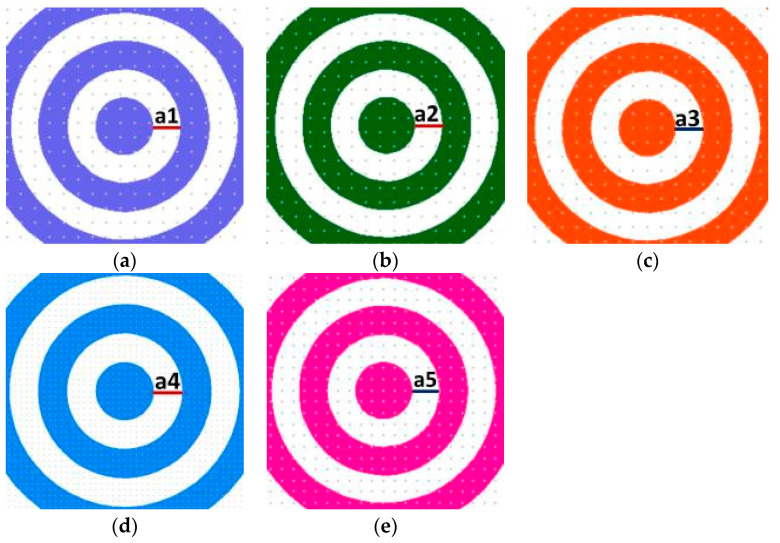
Photolithographic masks for multilevel axicon of order zero (**a**–**e**) M1–M5 for 5 photolithographic processes with different ring widths: (**a**) a1 = 160 μm, (**b**) a2 = 80 μm, (**c**) a3 = 40 μm, (**d**) a4 = 20 μm, (**e**) a5 = 10 μm.

**Figure 6 nanomaterials-13-00579-f006:**
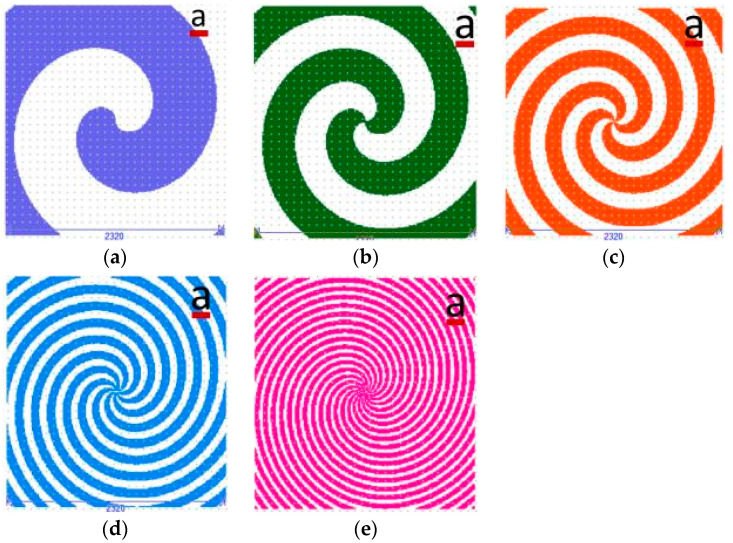
Photolithographic masks for multilevel axicon of order 1 (**a**–**e**) M1–M5 for five photolithographic processes.

**Figure 7 nanomaterials-13-00579-f007:**
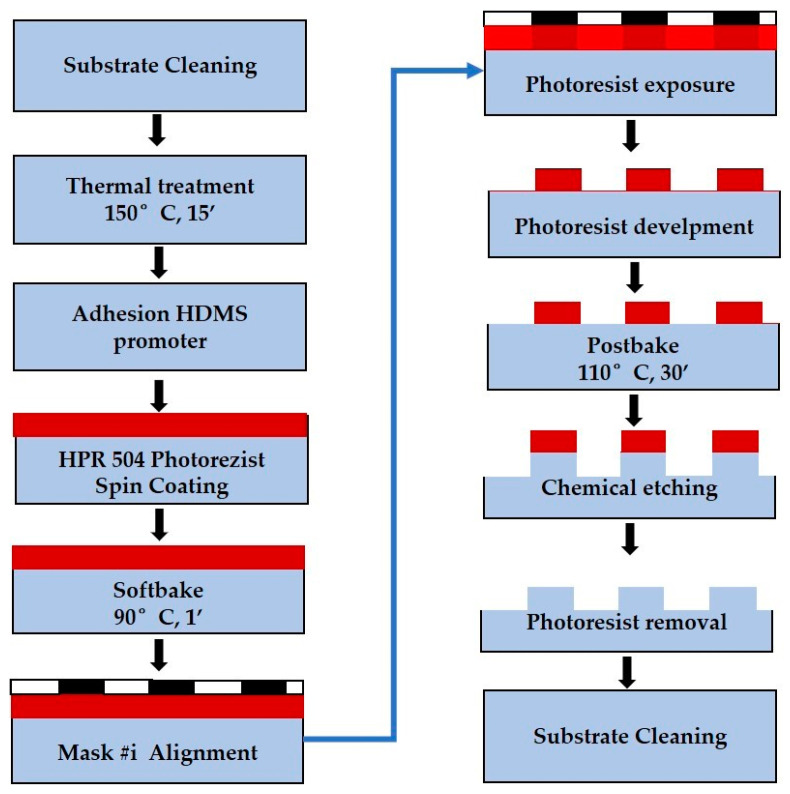
Flow chart for the fabrication of multilevel axicon with a number of **#i** photolithographic masks.

**Figure 8 nanomaterials-13-00579-f008:**
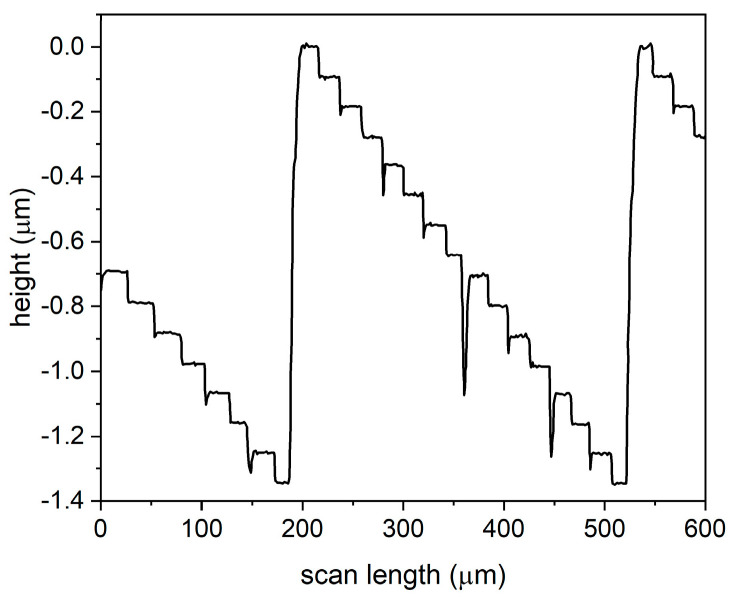
Cross-section profile of a zero-order axicon with 16 levels, measured with mechanical profilometer.

**Figure 9 nanomaterials-13-00579-f009:**
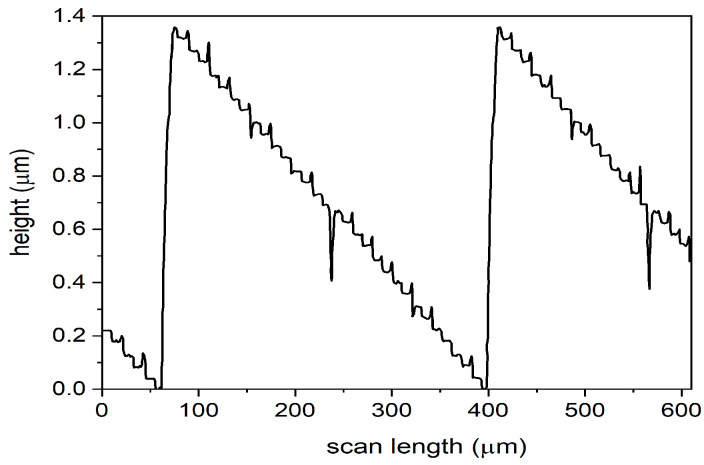
Cross-section profile of a zero-order axicon with 32 levels, measured with mechanical profilometer.

**Figure 10 nanomaterials-13-00579-f010:**
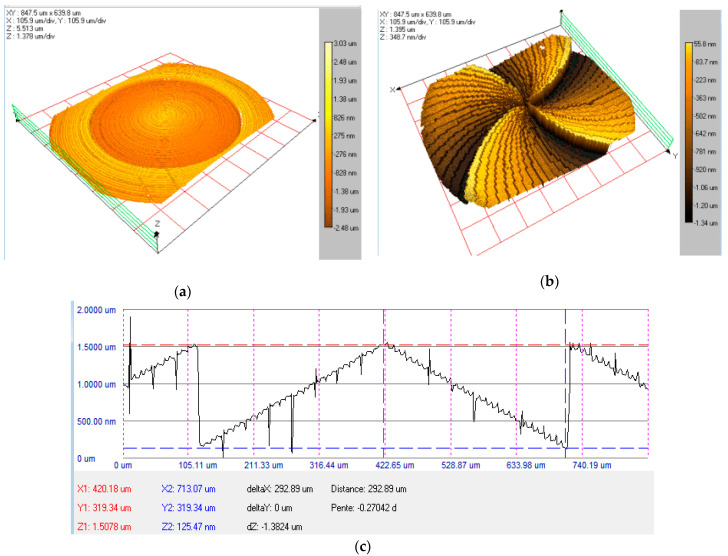
White light interferometry tridimensional maps for (**a**) zero-order axicon with 32 levels in fused silica, (**b**) spiral axicon with 32 levels in fused silica, (**c**) cross-section profile for zero-order axicon with 32 levels from (**a**).

**Figure 11 nanomaterials-13-00579-f011:**
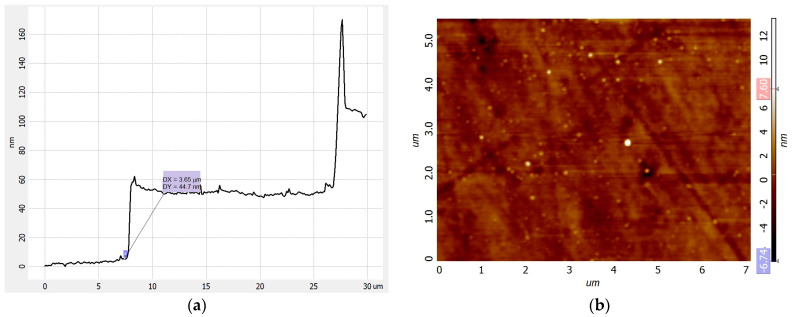
(**a**) AFM cross-section image for 32-level spiral axicon *m* = 4 in fused silica, 44.7 nm step, (**b**) roughness 2D profile—0.8 nm for 7 μm^2^ area.

**Figure 12 nanomaterials-13-00579-f012:**
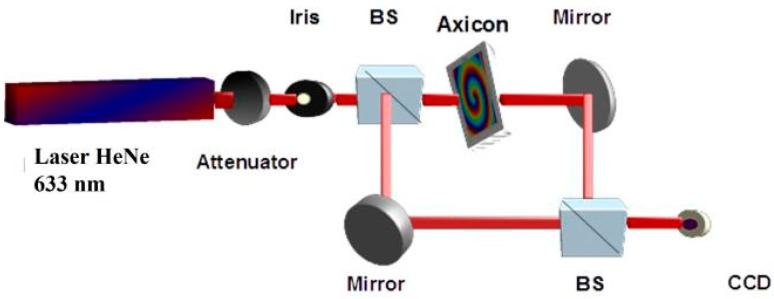
Optical setup for the generation of intensity and wavefront distribution for Bessel–Gauss beams.

**Figure 13 nanomaterials-13-00579-f013:**
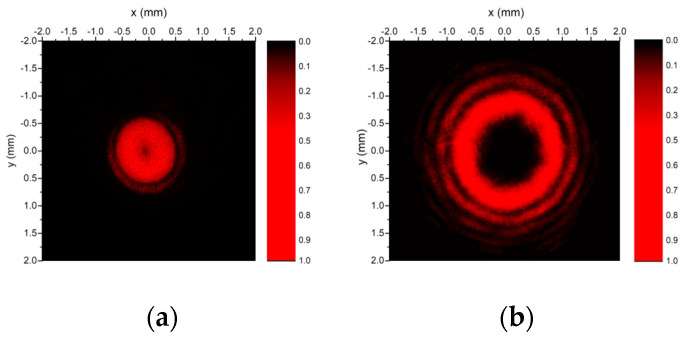
Measured intensity distribution of the beam generated by (**a**) a spiral axicon of order 1, and (**b**) a spiral axicon of order 4.

**Figure 14 nanomaterials-13-00579-f014:**
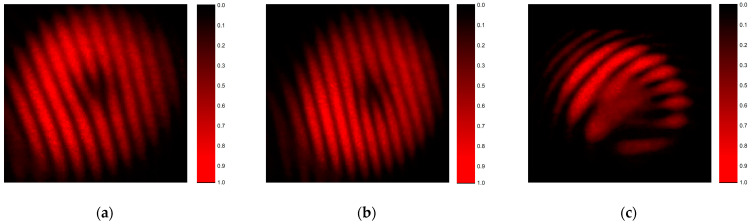
Fork interference patterns for beams generated by higher-order axicons with (**a**) *m* = 1, (**b**) *m* = −1, (**c**) *m* = −4.

**Figure 15 nanomaterials-13-00579-f015:**
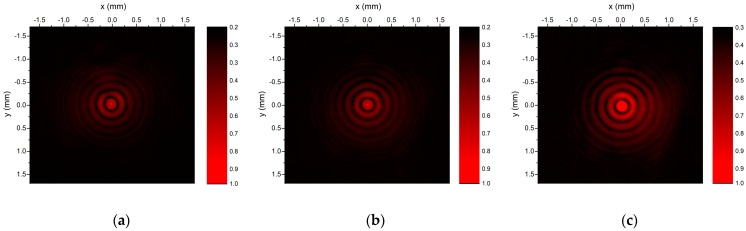
The 2D plots of measured intensities of zero-order Bessel–Gauss beam generated with zero-order axicon recorded at (**a**) z = 30 cm, (**b**) z = 37 cm, (**c**) z = 42 cm.

**Figure 16 nanomaterials-13-00579-f016:**
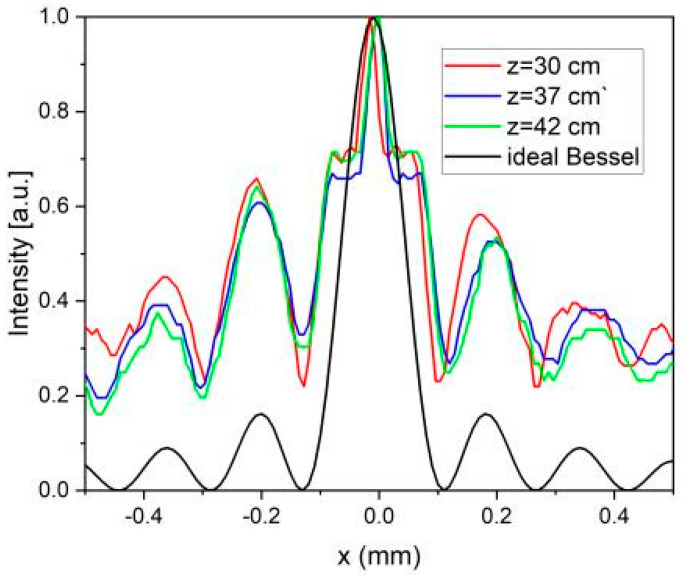
Intensity cross-sections of the zeroth Bessel-Gauss beams for z = 30 cm (red line), z = 37 cm (blue line), and z = 42 cm (green line). With the black line is represented the intensity of an ideal Bessel beam of order zero, |J_0_(k_r_x)|^2^.

**Figure 17 nanomaterials-13-00579-f017:**
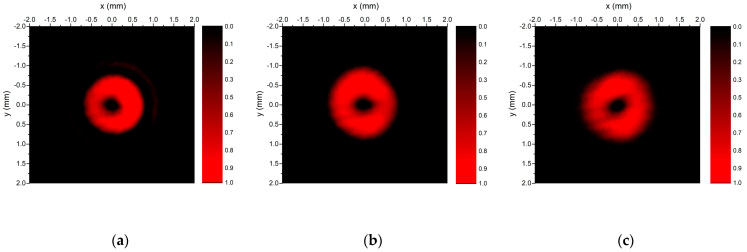
2D plots of measured intensities of zero-order Bessel–Gauss beam generated with zero-order axicon recorded at (**a**) z = 30 cm, (**b**) z = 40 cm, (**c**) z = 50 cm.

**Figure 18 nanomaterials-13-00579-f018:**
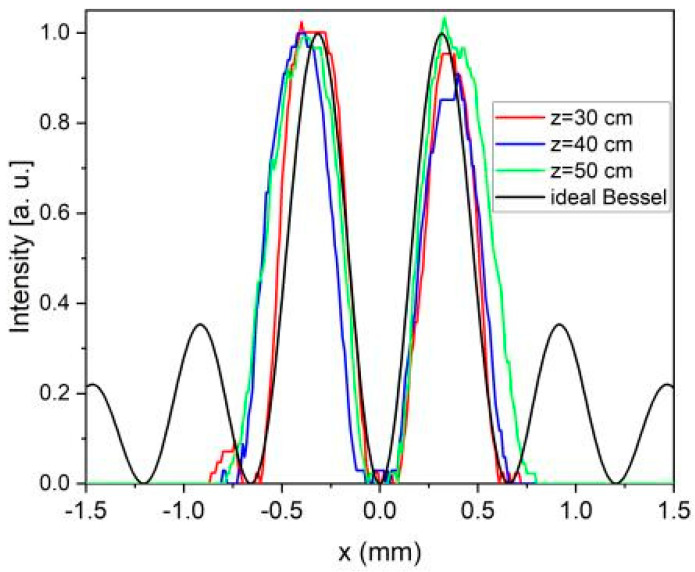
Intensity cross-sections of the first-order Bessel–Gauss beams for z = 30 cm (red line), z = 37 cm (blue line), and z = 42 cm (green line). The black line represents the intensity of an ideal Bessel beam of order zero, |J_1_(k_r_x)|^2^.

## Data Availability

Not applicable.

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
