# Peer review of "Multilevel Spiral Axicon for High-Order Bessel–Gauss Beams Generation"

_nanomaterials, 2023, doi:10.3390/nano13030579_

Round 1

Reviewer 1 Report

In the paper entitled “Multilevel spiral axicon for high-order quasi-Bessel beams generation”, the authors present the design, fabrication, and optical characterization of multilevel diffractive optics fabricated via multi-step etching of a glass substrate using different masks. The phase pattern encodes the combination of an axicon term with an azimuthal phase gradient in order to generate high-order Bessel-like beams. The matter is considered in depth from the three complementary aspects of design, fabrication, and optical characterization. The results support the claimed performance of the fabricated optical elements, and the paper can be of interest to the broad readership of the journal working in the field of nanofabrication and beam shaping. The paper can be accepted, provided the following points are addressed and discussed:

-        First paragraph: 30 years have passed since the seminal paper of Allen & coworkers

-        In the second paragraph, we suggest also mentioning conformal transformation among the methods for optical vortices generation, see for instance https://opg.optica.org/oe/fulltext.cfm?uri=oe-25-7-7859&id=362433 and https://opg.optica.org/oe/fulltext.cfm?uri=oe-27-4-3920&id=404658 and https://opg.optica.org/ol/abstract.cfm?uri=ol-47-14-3491

-        Throughout the paper, the authors refer to the beams generated by an axicon illuminated by a Gaussian beam as ‘quasi-Bessel’ beams. On the other hand, such beams are usually referred to as ‘Bessel-Gaussian' beams, see for instance https://opg.optica.org/ao/abstract.cfm?uri=ao-57-23-6725 and https://www.mdpi.com/2076-3417/10/21/7911 . Please consider this point and, in case, amend the definition throughout the paper using the more well-established one.

-        Please increase the size of Figures 1 and 2 and the font size of the labels and axes in order to make them more readable.

-        Figure 3: (a.u.) instead of (u.a.) on the y axes. We suggest centering the origin of the x-axis on the optical axis. In the present figure, the centre of the beam is at 2000 um. Moreover, in order to compare the experimental cross-section with the expected one, we suggest overlapping the theoretical curves so to make the readers appreciate the coincidence with the theoretical first minimum, as discussed in the text.

-        Equations at lines 110-111 are not numbered. Please put them inline or renumber all the equations accordingly.

-        Line 127: the diffractive optics introduce modulo 2pi phase shifts.

-        In Eq. (7), the number L of total levels seems not to have been introduced before.

-        Eq. (8) is valid if the whole element is illuminated, otherwise on the numerator we should have the product between R and the beam waist of the input beam. Please comment on this point.

-        Please introduce a scale bar in Figures 5 and 6. In particular, the differences between subfigures from (a) to (e) in Fig.5 are not clear. Please describe and make the difference more evident.

-        Line 221: the correct name of HMDS is hexamethyldisilazane

-        In Figures 7 and 8, anomalous peaks and dips appear, especially at the borders between different levels. Please comment on the origin of such deviations and on their effect on the quality of the output beam.

-        Please check Figure 11 in order to eliminate the red wavy lines below some words.

-        A scale bar should be added to Figures 12, 13, and 14. In addition, why Figures 13(a) and 13(b) have different color maps?

-        Finally, while the technique is effective for the fabrication of 3D diffractive optics in fused silica, on the other hand it is not clear the advantage with respect to maskless techniques, such as optical lithography or electron beam lithography (e.g. https://opg.optica.org/oe/fulltext.cfm?uri=oe-27-17-24123&id=416624 ) of a resist film deposited on the substrate. Please comment on this point and discuss it in the manuscript in order to further support the technique. 

Author Response

First of all we would like to thank the distinguished referee for his comments, questions and criticism to our manuscript. We did our best to address all this issues hoping to obtain an improved version of our work. In the following we provide our answers.

In the paper entitled “Multilevel spiral axicon for high-order quasi-Bessel beams generation”, the authors present the design, fabrication, and optical characterization of multilevel diffractive optics fabricated via multi-step etching of a glass substrate using different masks. The phase pattern encodes the combination of an axicon term with an azimuthal phase gradient in order to generate high-order Bessel-like beams. The matter is considered in depth from the three complementary aspects of design, fabrication, and optical characterization. The results support the claimed performance of the fabricated optical elements, and the paper can be of interest to the broad readership of the journal working in the field of nanofabrication and beam shaping. The paper can be accepted, provided the following points are addressed and discussed:

-        First paragraph: 30 years have passed since the seminal paper of Allen & coworkers

Answer: Introduction has been improved, see lines 25-29

-        In the second paragraph, we suggest also mentioning conformal transformation among the methods for optical vortices generation, see for instance https://opg.optica.org/oe/fulltext.cfm?uri=oe-25-7-7859&id=362433 and https://opg.optica.org/oe/fulltext.cfm?uri=oe-27-4-3920&id=404658  and https://opg.optica.org/ol/abstract.cfm?uri=ol-47-14-3491

Answer: As  you suggested, we introduced new references among the methods for optical vortices generation such as conformal transformation [23-25], but also metasurfaces [26, 27]. (line 42)

-        Throughout the paper, the authors refer to the beams generated by an axicon illuminated by a Gaussian beam as ‘quasi-Bessel’ beams. On the other hand, such beams are usually referred to as ‘Bessel-Gaussian' beams, see for instance https://opg.optica.org/ao/abstract.cfm?uri=ao-57-23-6725  and https://www.mdpi.com/2076-3417/10/21/7911. Please consider this point and, in case, amend the definition throughout the paper using the more well-established one.

Answer: The term ‘quasi-Bessel beam’ was replaced with the well established term ‘Bessel-Gauss’ beam.

-        Please increase the size of Figures 1 and 2 and the font size of the labels and axes in order to make them more readable.

Answer: Done. (Cross-section of the numerical intensities for the orders m=0, and m=1 for different propagation distances are represented  on the same plots, in order to illustrate the divergence-free behavior of Bessel beams and to allow a comparison with the experimental data see the newly introduced  Fig. 16 and Fig. 18.

-        Figure 3: (a.u.) instead of (u.a.) on the y axes. We suggest centering the origin of the x-axis on the optical axis. In the present figure, the centre of the beam is at 2000 um. Moreover, in order to compare the experimental cross-section with the expected one, we suggest overlapping the theoretical curves so to make the readers appreciate the coincidence with the theoretical first minimum, as discussed in the text.

Answer: Done

-        Equations at lines 110-111 are not numbered. Please put them inline or renumber all the equations accordingly.

Answer: Equations at lines 110-111 (now at lines 116, 118) are numbered and in-line. All the equations are renumbered accordingly.

-        Line 127: the diffractive optics introduce modulo 2pi phase shifts.

Answer: Line 127, (now 134) corrected as you suggested: (diffractive axicons) introduce modulo 2pi phase shifts.

-        In Eq. (7), the number L of total levels seems not to have been introduced before.

Answer: Number L of total levels is introduced now at line 143 and used in Eq (7), now Eq (12) 

-        Eq. (8) is valid if the whole element is illuminated, otherwise on the numerator we should have the product between R and the beam waist of the input beam. Please comment on this point.

Answer: Done. An appropriate comment was introduced in the text at  the former equation 8 now equation 11.

-        Please introduce a scale bar in Figures 5 and 6. In particular, the differences between subfigures from (a) to (e) in Fig.5 are not clear. Please describe and make the difference more evident.

Answer: Figures 5 and 6 have scale bar, now, in order to clarify the differences between subfigures from (a) to (e).

-        Line 221: the correct name of HMDS is hexamethyldisilazane

Answer: Line 221, now 240, the corrected name of HMDS is hexamethyldisilazane

-        In Figures 7 and 8, anomalous peaks and dips appear, especially at the borders between different levels. Please comment on the origin of such deviations and on their effect on the quality of the output beam.

Answer: In Figures 7 and 8, Now Fig 8, 9  anomalous peaks and dips appear, especially at the borders between different levels because of photolithographical masks alignment. Since these deviations are less than 1 µm, and the area of fabricated axicons is much larger (4x4=16 mm2 in case of zero-order axicon or 2.3 x2.3 mm2 in case spiral axicons), the effect of the peaks and dips is negligible.

-        Please check Figure 11 in order to eliminate the red wavy lines below some words.

Answer: We checked Figure 11 (now Fig 12) in order to eliminate the red wavy lines below some words.

-        A scale bar should be added to Figures 12, 13, and 14. In addition, why Figures 13(a) and 13(b) have different color maps?

Answer: Figures 12, 13, and 14 (NOW Figures 13, 14, and 15) have now scale bar, the same color maps.

 -        Finally, while the technique is effective for the fabrication of 3D diffractive optics in fused silica, on the other hand it is not clear the advantage with respect to maskless techniques, such as optical lithography or electron beam lithography (e.g. https://opg.optica.org/oe/fulltext.cfm?uri=oe-27-17-24123&id=416624) of a resist film deposited on the substrate. Please comment on this point and discuss it in the manuscript in order to further support the technique. 

Answer: See lines:62-75

Author Response

First of all we would like to thank the distinguished referee for his comments, questions and criticism to our manuscript. We did our best to address all this issues hoping to obtain an improved version of our work. In the following we provide our answers.

The manuscript presents a method for the generation of high-order quasi-Bessel beams carrying orbital angular momentum using a multilevel spiral axicon. The fabrication process and characterization of the multilevel spiral axicon is demonstrated. Some major concerns regarding the manuscript are,

1) The manuscript does not provide a clear novelty of the proposed work in comparison to the already existing works. Therefore, the authors need to clarify the novelty of the method by comparing the advantages of this method and fabrication by comparing with various existing methods for the generation of diffraction-free optical vortex beams.

Answer: We do believe that the relevance of this work consists of the application of the microfabrication technologies more specifically the photolithography followed by the chemical etching in order to realize optical/micro-optical components with free form optical surfaces and various diameters ranging from hundreds of µm to tens or hundreds of millimeters. While for free form optical surfaces, various sophisticated techniques such as single point diamond turning machining [1,2], molding [3] or 3D lithography [4] are applied, they present several challenges in terms of complexity, costs, and diameters of the components to be fabricated. Microfabrication techniques represent a mature technology, a standard in micro/nanoelectronics presenting the advantages of low cost, mass production and flexibility in terms of the wafer diameters. It is the aim of this work to develop and establish a technological flow using simple technological processes for the realization of optical components with aspherical surfaces, specifically, axicon and spiral axicons. This paragraph was added into the introductory section (lines 62 – 75) as well as the  references [31-37].

2) The authors need to address their previous work “Multilevel axicon for perfect optical vortex generation” and highlight the advantages of the current method over the previous one.

In their previous work, “Multilevel axicon for perfect optical vortex generation”, the authors presented the fabrication of zero-order axicons working in reflection mode for visible light (λ=633 nm). There are several advantages of the current method such as:

  • Small roughness of the surface (roughness 0.8 nm for 7 μm2) in case of fused silica axicon
  • Simplified optical setup via transmission mode (at λ=633 nm) of the axicon fabricated in fused silica material which reduces the errors of alignment and illumination at an angle, different than normal incidence in case of the reflective axicons.

3) The characterization of the fabricated multilevel spiral axicon using a Mach Zehnder interferometer is demonstrated with experimental results.

  1. a) Why the authors didn’t demonstrate the reconstructed amplitude and phase distributions of generated OAM modes (fork interference pattern is available)?

The phase distribution will give the direct visualization of the sign and magnitude of the topological charge of OAM modes.

Answer: The fork interference patterns are available for OAM modes m=1 (Figure 13c), m=-1 (Figure 13d) and m=4  (Figure 14c). These fork fringes represent the wavefront characterization of the Bessel beam modes and illustrates the sign (see m=1 (Figure 13c), m=-1 (Figure 12d)) and the magnitude of the topological charge of OAM modes (one fringe in case of OAM m=1 and 4 fringes in case of OAM m=4).  

  1. b) All experimental results require the respective intensity map and scale bar.

Answer: Done. We plotted the figures with reference size, intensity map, and scale bars. The interference figures represented in Fig. 13 do not have a reference size since their role is simply illustrative, they put in evidence the fork – like dislocations a signature of the phase singularity, the helical character of the wavefronts and the topological charges.

  1. c) Why Fig.13 (a) and (b) looks different in representation? More discussions required in these figures

Answer: Fig.13 (a) and (b)    look different in representation because they are recorded with different CCD cameras. Now the images from Fig 13 are similar in representation.

4) A comparison of theoretical results and experimental results (from fabricated multilevel spiral axicon) with proper profile plots (showing divergence-free behavior) may give a better understanding of the efficiency of the fabricated spiral axicons.

Answer: A comparison of theoretical results and experimental results (from fabricated multilevel spiral axicon) with proper profile plots (showing divergence-free behavior) was done. Figs 16 and 18 show the cross sections of the 2D intensity maps is shown in Figs. 15 and 17.

5) The discussion part (section 4) does not properly convey the significance and applicability of the proposed method. More clear and detailed discussion is required in this section.

Answer: We consider that this fabrication method, besides the advantages mentioned in the introductory section (lines 62 – 75), presents the potential to fabricate diffractive optical elements with larger diameters up to the size of the wafer. In the case of axicons, large diameters will allow the possibility to obtain longer distances in which the generated beam is non – diffractive This comment was introduced in the conclusions lines 448 - 452.  

Reviewer 3 Report

In their manuscript “Multilevel spiral axicon for high-order quasi-Bessel beams generation”, the authors propose to generate quasi-Bessel beams with different orders by multi-level spiral axicons and present the fabrication of the proposed axicons. Axicons with 16 and 32 levels are proposed, which work at the wavelength of 633 nm, the results show that the axicons can generate different orders quasi-Bessel beams efficiently. In my opinion, this paper can be accepted for publication after a revision by considering the following issues:

1.      The picture and word texts in Figures 1-3, 5 are too small for reading. The results of Figure 3 (a) and (b) ((c) and (d)) can be integrated into a figure, it can compare the intensity result on different z positions intuitively.

2.      There are some grammar and description mistakes in the manuscript, for example, on page 3 line 99, “proportional with” should revise to “proportional to”, and the figure caption of Figure 9 seems incorrect. Please check and revise the mistakes carefully.

3.      Equation (10) on page 4 line 154 should be displayed on an alone line. The description on page 4 lines 159-162 is ambiguous, please revise it clearly according to Figure 5. Please clarify the definition of “conversion efficiency” on page 6 line 241.

4.      Section 3.2 describe the fabrication of the multilevel axicons in detail, it’s recommended to show a corresponding flow chart for better comprehension.

5.      The results of Figures 12-15 lack reference to the size. To highlight the divergence-free property of the generated Bessel beam, it’s recommended to analyze the size of the foci or hollow at different distances.

6.      The manuscript proposes the spiral axicons with 16 and 32 levels, but there are no comparisons between them, what are the main differences between them?

7.      The methodology of metasurfaces seems to be helpful for the generation of bessel beam,  and some related works migh be helpful, for example (a)  doi: 10.1021/acsami.2c00742, (b)  doi: 10.1002/lpor.202200777

Author Response

First of all we would like to thank the distinguished referee for his comments, questions and criticism to our manuscript. We did our best to address all this issues hoping to obtain an improved version of our work. In the following we provide our answers.

In their manuscript “Multilevel spiral axicon for high-order quasi-Bessel beams generation”, the authors propose to generate quasi-Bessel beams with different orders by multi-level spiral axicons and present the fabrication of the proposed axicons. Axicons with 16 and 32 levels are proposed, which work at the wavelength of 633 nm, the results show that the axicons can generate different orders quasi-Bessel beams efficiently. In my opinion, this paper can be accepted for publication after a revision by considering the following issues:

  1. The picture and word texts in Figures 1-3, 5 are too small for reading. The results of Figure 3 (a) and (b) ((c) and (d)) can be integrated into a figure, it can compare the intensity result on different z positions intuitively.

Answer: We resized Figures 1-3, 5 and the text. The results of Figure 3 (a) and (b) ((c) and (d)) are now integrated into a figure in order to compare the intensity result on different z positions.

  1. There are some grammar and description mistakes in the manuscript, for example, on page 3 line 99, “proportional with” should revise to “proportional to”, and the figure caption of Figure 9 seems incorrect. Please check and revise the mistakes carefully.

Answer: On page 3 line 99 (now 103), “proportional with” was revised to “proportional to”. Figure caption of Figure 9 has been corrected.

  1. Equation (10) on page 4 line 154 should be displayed on an alone line. The description on page 4 lines 159-162 is ambiguous, please revise it clearly according to Figure 5. Please clarify the definition of “conversion efficiency” on page 6 line 241.

Answer: Equation (10)  now equation (13) on page 4 line 154 (line 160 now) are displayed on an alone line.

  1. Section 3.2 describe the fabrication of the multilevel axicons in detail, it’s recommended to show a corresponding flow chart for better comprehension.

Answer: The fabrication of the multilevel axicons has a corresponding flow chart for better comprehension.

  1. The results of Figures 12-15 lack reference to the size. To highlight the divergence-free property of the generated Bessel beam, it’s recommended to analyze the size of the foci or hollow at different distances.

Answer: Figures 12-15, Now Figures 13-16 have reference to the size. To highlight the divergence-free property of the generated Bessel beam, it was analyzed the size of the foci or hollow at different distances by introducing the new figures Fig 16 and Fig 18 representing the comparison between the cross sections of the 2D plots represented in Fig. 15 and Fig 17. For comparison we overlayed on Figs 16 and 18 the intensity of the ideal Bessel beams. Appropriate explanations were introduced in the text lines 390 -434.

  1. The manuscript proposes the spiral axicons with 16 and 32 levels, but there are no comparisons between them, what are the main differences between them?

Answer: Indeed the manuscript describes the technological processes for axicons with 16 and 32 levels. Nevertheless the axicons with 16 levels were simply an intermediary step for calibrating the technological processes. In the experimental characterizations we disscussed only the 32 level axicons since they do have a better calculated efficiency, better surface approximation of a continuous profile which allows a better OAM purity.

  1. The methodology of metasurfaces seems to be helpful for the generation of bessel beam,  and some related works migh be helpful, for example (a)  doi: 10.1021/acsami.2c00742, (b)  doi: 10.1002/lpor.202200777

The recommended references in case of the methodology of metasurfaces for the generation of bessel beam are added [26, 27].

Reviewer 4 Report

The authors have written an importnat paper from a practical point of view devoted to fabrication of thin and compact multilevel axicon for generation of high order quasi Bessel beams with orbital angular momentum. The paper is written quite clearly and will be useful for many specialists in photonics. I believe it can be published in Nanomaterials provided that several questions that arose during the reading are clarified.

1. On line 89 and below, the authors write first about the Bessel beam of zero order and then about spiral axicon of order of 1. As far I understand this is the same name of the Bessel beam, so it probably makes sense to use one name for them.

2. It seems to me thath the authors need to explain what is the diffraction error and diffraction depth on line 154 (what is epsilon).

3. In formula 9 the argument has dimension it is apparently worth correcting. 

Author Response

First of all we would like to thank the distinguished referee for his coments, questions and criticism to our manuscript. We did our best to address all this issues hoping to obtain an improved version of our work. In the following we provide our answers.

The authors have written an importnat paper from a practical point of view devoted to fabrication of thin and compact multilevel axicon for generation of high order quasi Bessel beams with orbital angular momentum. The paper is written quite clearly and will be useful for many specialists in photonics. I believe it can be published in Nanomaterials provided that several questions that arose during the reading are clarified.

  1. On line 89 and below, the authors write first about the Bessel beam of zero order and then about spiral axicon of order of 1. As far I understand this is the same name of the Bessel beam, so it probably makes sense to use one name for them.

Answer: Done.

  1. It seems to me thath the authors need to explain what is the diffraction error and diffraction depth on line 154 (what is epsilon).

Answer: We deleted the formula for diffraction errror, we calculate the conversion efficiency of multilevel axicon with L levels according to Equation (12). The errors of fabrication depths are calculated in section Structural charactherization.

  1. In formula 9 the argument has dimension it is apparently worth correcting. 

Answer: In formula 9, now 12, from reference [38] – page 30,31,  the argument  L is adimensional, representing the number of levels of the fabricated axicon.

Round 2

Reviewer 1 Report

The authors have properly addressed all the points and issues raised during the review. The manuscript has been significantly improved and can be now accepted for publication.

Reviewer 2 Report

In the revised version of the manuscript, the authors have complied with my questions or suggestions in a satisfactory way. I therefore, feels the manuscript can be considered for publication in Nanomaterials.